# Cytogenetic Analysis of Sporadic First-Trimester Miscarriage Specimens Using Karyotyping and QF-PCR: A Retrospective Romanian Cohort Study

**DOI:** 10.3390/genes13122246

**Published:** 2022-11-29

**Authors:** Gabriela Popescu-Hobeanu, Anca-Lelia Riza, Ioana Streață, Ștefania Tudorache, Alexandru Comănescu, Florentina Tănase, Roxana Cristina Drăgușin, Cornelia Pascu, Anda Lorena Dijmărescu, Monica-Laura Cara, Ștefania Dorobanțu, Bianca Petre-Mandache, Mihai Cucu, Simona Serban Sosoi, Mihai Ioana, Dominic Iliescu, Florin Burada

**Affiliations:** 1Doctoral School, University of Medicine and Pharmacy of Craiova, 200349 Craiova, Romania; 2Laboratory of Human Genomics, University of Medicine and Pharmacy of Craiova, 200638 Craiova, Romania; 3Regional Centre of Medical Genetics Dolj, Emergency Clinical County Hospital Craiova, 200642 Craiova, Romania; 4Department of Obstetrics and Gynecology, University of Medicine and Pharmacy of Craiova, 200349 Craiova, Romania; 5Department of Obstetrics and Gynecology, Emergency Clinical County Hospital, 200642 Craiova, Romania; 6Gyn Med Clinic, 200074 Craiova, Romania; 7Department of Obstetrics and Gynecology, Clinical Municipal Hospital “Filantropia” of Craiova, 200143 Craiova, Romania; 8Department of Public Health, University of Medicine and Pharmacy of Craiova, 200349 Craiova, Romania

**Keywords:** first-trimester miscarriage, chromosome abnormality, karyotype, QF-PCR, maternal cell contamination

## Abstract

It is well known that first-trimester miscarriages are associated with chromosome abnormalities, with numerical chromosome abnormalities being the ones most commonly detected. Conventional karyotyping is still considered the gold standard in the analysis of products of conception, despite the extended use of molecular genetic techniques. However, conventional karyotyping is a laborious and time-consuming method, with a limited resolution of 5–10 Mb and hampered by maternal cell contamination and culture failure. The aim of our study was to assess the type and frequency of chromosomal abnormalities detected by conventional karyotyping in specimens of sporadic first-trimester miscarriages in a Romanian cohort, using QF-PCR to exclude maternal cell contamination. Long-term cultures were established and standard protocols were applied for cell harvesting, slide preparation, and GTG banding. All samples with 46,XX karyotype were tested for maternal cell contamination by QF-PCR, comparing multiple microsatellite markers in maternal blood with cell culture and tissue samples. Out of the initial 311 specimens collected from patients with sporadic first-trimester miscarriages, a total of 230 samples were successfully analyzed after the exclusion of 81 specimens based on unsuitable sampling, culture failure, or QF-PCR-proven maternal cell contamination. Chromosome abnormalities were detected in 135 cases (58.7%), with the most common type being single autosomal trisomy (71/135—52.6%), followed by monosomy (monosomy X being the only one detected, 24/135—17.8%), and polyploidy (23/135—17.0%). The subgroup analysis based on maternal age showed a statistically significant higher rate of single trisomy for women aged 35 years or older (40.3%) compared to the young maternal age group (26.1%) (*p* = 0.029). In conclusion, the combination of conventional karyotyping and QF-PCR can lead to an increased chromosome abnormality detection rate in first-trimester miscarriages. Our study provides reliable information for the genetic counseling of patients with first-trimester miscarriages, and further large-scale studies using different genetic techniques are required.

## 1. Introduction

Human reproduction is an inefficient process, with an estimated 40–60% of all conceptions failing between fertilization and birth [1]. About 10–15% of all clinical pregnancies end in early spontaneous miscarriage [2], with the most common cause being sporadic chromosomal abnormalities [3]. Non-genetic causes of pregnancy loss include but are not limited to congenital structural uterine anomalies or fibroids [4]; endocrine disorders (polycystic ovary syndrome, poorly controlled diabetes mellitus, and various thyroid disorders) or thrombophilic factors [5,6]; certain autoimmune disorders, including the presence of autoantibodies directed against various cells and tissues [7,8]; high-fever infections [9,10]; and various lifestyle factors [11,12].

It has been reported that around half of early spontaneous abortions seem to randomly occur [13,14] and are due mainly to de novo meiotic errors, although mitotic or fertilization errors, or unbalanced genomic rearrangements can also occur [15,16]. The most common chromosome abnormalities detected in spontaneous miscarriages are aneuploidies (autosomal trisomies and monosomy X) and polyploidies (triploidies and tetraploidies) [17,18,19,20,21].

Chromosomal non-disjunction in maternal meiosis is the main cause of autosomal trisomy, leading to an increased frequency of viable trisomies (Patau, Edwards, and Down syndromes) in advanced maternal age pregnancies [22,23,24]. On the other hand, monosomy X is caused by paternal meiotic errors in more than 50% of cases [25,26]. Mitotic errors result in multiple cell lines (mosaicism) [27], whereas fertilization errors lead to triploidy, either digyny (the extra set of chromosomes is maternal in origin) or diandry (paternal). Tetraploidy is mainly caused by cytokinesis failure during the first post-zygotic mitosis, with the result being an abnormal cell with four haploid sets [3].

Advanced maternal age is related to an increased incidence of meiotic errors due to the extended period of oocyte meiotic arrest before ovulation [28,29]. The risk of cytogenetic errors was reported to be significantly higher in women over 40 years old [30,31]. There seems to be conflicting data on the frequency of chromosome abnormalities in sporadic miscarriages, as opposed to recurrent pregnancy loss (RPL), with some reports claiming that such abnormalities occur less often in RPL, and others reporting no correlation [32,33,34,35,36].

Testing specimens is critically important for assessing the genetics of pregnancy loss. Standard karyotyping is still considered the gold standard in the analysis of products of conception due to its ability to detect all numerical abnormalities, and large unbalanced and balanced structural rearrangements; its relatively low cost; and its high accessibility [19,37]. However, conventional karyotyping is a highly laborious and time-consuming method, with a limited resolution of 5–10 Mb and hampered by maternal cell contamination and culture failure [38]. Recently, we witnessed the spread of a wide variety of complementary molecular techniques available for the analysis of product-of-conception analyses, including QF-PCR, MLPA, FISH, and chromosomal microarray (CMA) [39,40].

To our knowledge, data on the genetics of early miscarriage obtained using both karyotyping and molecular methods is currently scarce. The purpose of our study was to investigate the frequency and type of chromosomal abnormalities detected by standard karyotyping following maternal cell contamination exclusion by QF-PCR in sporadic spontaneous first-trimester miscarriages in a southwestern Romanian cohort. A stratified analysis based on maternal age was also performed.

## 2. Materials and Methods

The product-of-conception samples included in this study were referred to the Human Genomics Laboratory of Craiova, Romania for cytogenetic analysis between January 2013 and August 2022, by the Department of Obstetrics and Gynecology of the Emergency Clinical County Hospital, the Department of Obstetrics and Gynecology of the Clinical Municipal Hospital “Filantropia” of Craiova and Gyn Med Clinic. Data concerning reproductive history, including maternal and gestational ages, were recorded. All cases diagnosed with recurrent pregnancy loss were excluded in this study. The study was conducted in accordance with the Declaration of Helsinki and approved by the Ethics Committee of the University of Medicine and Pharmacy of Craiova, Romania (no. 44/24.03.2022).

### 2.1. Conventional Cytogenetic Analysis

A chromosome analysis of the specimens was performed following long-term culture according to standard cytogenetic methods using the G-banding technique. Each specimen delivered to the laboratory was immediately placed in a sterile Petri dish, rinsed using fresh RPMI 1640 Medium, and dissected from blood clots and endometrial tissue. Long-term cell cultures (7–14 days) were established from the processed fetal tissue using AmnioMAX™ Complete Medium (Gibco Invitrogen, Waltham, MA, USA) in T-25 cell culture flasks at 37 °C and 5% CO_2_, according to standard protocols. The cells were then arrested in metaphase with colcemid, trypsinized, processed using hypotonic solution and methanol–acetic acid 3:1 fixative, and spread on microscope slides, which were subsequently G-banded. At least 20 metaphases were analyzed and karyotyped for each sample (Ikaros v5.4, MetaSystems, Altlussheim, Germany). The karyotypes were reported according to ISCN 2013, 2016, and 2020.

### 2.2. Quantitative Fluorescent Polymerase Chain Reaction (QF-PCR)

QF-PCR is a method used for detecting chromosome copy number by amplification of polymorphic DNA markers (short tandem repeats) specific to the chromosomes of interest (commonly 13, 18, 21, X, and Y). Following PCR using fluorescent primers, the amplification products are separated by size using a capillary electrophoresis system, and the amount of DNA present in each fragment is automatically quantified.

All samples with a normal female karyotype were subjected to QF-PCR testing for maternal cell contamination by comparison of multiple microsatellite markers in maternal blood versus cell culture and/or fetal sample. The IVD QF-PCR Devyser (Devyser AB, Stockholm, Sweden) prenatal kit, testing for aneuploidies of chromosomes 21, 18, 13, X, and Y, as well as the extended kit for chromosomes 15, 16, and 22 were used. DNA purification was performed using Promega Wizard™ Genomic (Promega, Madison, WI, USA). The amplicons were migrated on the ABI3730xl platform (Applied Biosystems, Foster City, CA, USA), and data were analyzed using GeneMarker v2.2 software (SoftGenetics, State College, PA, USA). 

### 2.3. Statistical Analysis

Data were analyzed by using SPSS Statistics for Windows, Version 22.0 (IBM SPSS Statistics for Windows, Version 22.0. Armonk, NY, USA: IBM Corp).

Descriptive statistics were produced for all study variables. Continuous variables are presented as the mean ± SD, and categorical data were summarized by frequency and per-centage. A stratified analysis based on maternal age (<35 years old and ≥35 years old) was performed using Χ^2^ test or Fisher’s exact test, with *p* < 0.05 being considered statistically significant.

## 3. Results

Out of the initial 311 samples collected from patients with sporadic first-trimester miscarriages, 81 were excluded based on unsuitable sampling, culture failure, or QF-PCR proved maternal cell contamination.

Normal (either 46,XX or 46,XY) karyotypes were observed in 95 cases (41.3%), while abnormal karyotypes were observed in 135 cases, resulting in a detection rate of 58.7% for chromosome abnormalities in first-trimester miscarriages. Table 1 provides a more detailed view of the chromosome abnormalities detected in the present study.

### 3.1. Aneuploidy

Single trisomy was the most frequently detected abnormality, accounting for 71 of the 135 cases (52.6%): 21 cases involved chromosome 13, 18, or 21; common non-viable trisomies involving chromosomes 15, 16, and 22 were observed in 38 cases; and 12 cases consisted of rare, non-viable trisomies involving other chromosomes. No sex chromosome single trisomies were detected. Trisomy 16 was the most common single trisomy (23 cases, 32.4% of single trisomies, 17% of all detected abnormalities), followed by trisomy 21 (10 cases, 14.1% of single trisomies, 7.4% of all abnormalities) and trisomy 22 (9 cases, 12.7% of single trisomies, 6.7% of all abnormalities). Table 2 shows the involvement of each autosome in the aforementioned single trisomies. Three cases of double trisomy were detected: one 48,XX, +6, +10, one 48,XXY, +13, and one 48,XYY, +22.

A pure 45,X karyotype was detected in 24 samples, representing 17.8% of all abnormal karyotypes. Monosomy X was the most prevalent single chromosome abnormality in the present series.

The mosaic category includes three cases that had multiple cell lines: mos 45,X/46,XY, mos 48,XX, +12, +13/46,XX, and mos 49,XX, +9, +12, +20/46,XX, together representing 2.2% of all chromosome abnormalities detected.

### 3.2. Polyploidy

Sixteen samples showed triploidy: ten cases with 69,XXX and six with 69,XXY. Pure triploidy was the most frequent single chromosome abnormality (11.9%), behind monosomy X (17.8%) and trisomy 16 (17%). Two cases showed triploidy with two extra autosomes: 71,XXY, +3, +20 and 71,XXY, +5, +9 (hypertriploidy).

Tetraploidy was observed in 3% of the abnormal karyotypes; there were two cases with 92,XXXX and two cases with 92,XXYY. One case showed tetraploidy with one extra autosome: 92,XXXX, +15 (hypertetraploidy).

### 3.3. Structural Abnormalities

Three cases showed an apparently balanced chromosome rearrangement: two pericentric inversions and one reciprocal translocation. Another pericentric inversion was observed with an additional trisomy and included in the single trisomy category. An unbalanced chromosome rearrangement occurred in eight cases (5.9%): four unbalanced Robertsonian translocations, two terminal deletions, one unbalanced reciprocal translocation, and one isochromosome (Table 3).

### 3.4. Chromosome Abnormalities and Maternal Age

Maternal age ranged between 18 and 46 years old. The patients were divided into two age groups: young maternal age, <35 years (YMA, *n* = 153), and advanced maternal age, ≥35 years (AMA, *n* = 77). The rate of chromosome abnormalities was higher in the AMA group (67.5%) compared to the YMA group (54.2%), without reaching statistical significance (*p* = 0.053). Upon investigating differences regarding abnormality types between groups, we found a statistically significant higher rate of single trisomy for the AMA group (40.3%) compared to the YMA group (26.1%) (*p* = 0.029). On the contrary, the YMA showed a higher rate of polyploidy (12.4%) compared to the AMA group (5.2%), but no significant difference was found (*p* = 0.085). The rates for monosomy X were similar between both groups (10.5% for YMA vs. 10.4% for AMA) (Table 4).

## 4. Discussion

Our 10-year study provides additional data on the frequency, distribution, and types of chromosomal abnormalities found in sporadic first-trimester miscarriages using QF-PCR to exclude maternal cell contamination.

We found an increased abnormality rate (58.7%) compared with most previous studies [14,19,38,41,42,43,44,45]. However, there are reported data in which the abnormality rate was higher [2,16,19,31,46]. Our findings compared to the relative frequencies of different types of chromosome abnormalities along with the ones reported in previous studies are shown in Table 5.

Single autosomal trisomies are the most common type of chromosome abnormality in first-trimester miscarriages. In our study, trisomy 16 was the most frequent one (17% of all abnormalities), followed by trisomy 21 (7.4%) and trisomy 22 (6.7%). Trisomy 18 (5.2%), trisomy 15 (4.4%), and trisomy 13 (3%) are the following most common single autosomal trisomies (Table 6).

In our case-series, monosomy X was the most common single chromosome abnormality (17.8%). This finding could be explained by the idea that lethality in 45,X cases is caused by the absence of rescue cell lines (46,XX) in critical tissues such as the placenta [47]. We found no statistically significant association between monosomy X and maternal age.

We found a significantly higher rate of single autosomal trisomy in women over the age of 35 (40.3%) compared to younger patients (26.1%) (*p* = 0.029). Only few stratified analyses on the correlation between the incidence of single autosomal trisomies and maternal age are available, and a direct comparison is difficult due to differently formatted data. However, there are some previous studies where a higher rate of single autosomal trisomy was observed in older women [16,19,46,48,49,50].

We also found a polyploidy rate of 17%, consistent with other studies [2,16,19,31,41]; 78.3% of polyploidies were triploidies, amounting to 13.3% of all detected chromosome abnormalities, with the YMA group showing a higher incidence of polyploidy (12.4%), compared to the AMA group (5.2%).

Even though most miscarriages are isolated events, understanding their etiology provides comfort to patients. Genetic testing of products of conception by karyotyping aids reproductive counseling either by providing a reason for fetal loss in cases when chromosomal abnormalities are detected or by redirecting the search towards other etiologies in cases with a normal karyotype. Genetic and epigenetic abnormalities undetectable by karyotyping may lead to early pregnancy loss. Potential genetic causes of miscarriage in euploid pregnancies include single-nucleotide variants that affect individual genes (changes in genes involved in centrosome integrity, anti-inflammatory and immune responses, cell differentiation and proliferation, blood coagulation, connective tissue, endocrine and neuromuscular systems, etc.) detectable by sequencing and submicroscopic abnormalities (inherited or de novo copy number variants involving genes associated with embryo implantation, growth and early development, immune signaling) detectable by microarray analysis [51,52]. The aberrant DNA methylation of several imprinted loci at genes implicated in oocyte development, maturation, skewed X chromosome inactivation, or histone modifications in placental tissue can also contribute to euploid miscarriage [51,53].

The strength of our study resides in the fact that all results were obtained using standardized procedures carried out in a single laboratory to a good cell culture success rate (over 75%) and that misdiagnosis due to maternal cell contamination was minimized by using molecular techniques (QF-PCR).

This study is limited by the small sample size. Conventional karyotyping of first-trimester product-of-conception samples has obvious limitations due to a high risk of culture failure. Recent studies have proven the fact that the type of tissue sample collected is critical to cell growth success and subsequent karyotype analysis, with placental villi being the highest (>80%) and fetal parts being the lowest (<40%) in terms of cell culture success [54,55]. There is a reported 10–40% cell growth failure determined by factors such as the period of time between the moment of the loss and the sample collection, and the bacterial or fungal contamination of the sample [56,57]. It has been suggested that abnormal chorionic villus cells may show impaired in vitro proliferation in long-term culture, leading to the underestimation of the abnormality rate [15]. Alternative techniques complementary to conventional karyotyping (QF-PCR, MLPA, FISH, array CGH, and SNP array) should also be used for genetic testing to overcome the issues of maternal cell contamination and limited resolution. Chromosomal rearrangements seem to be cryptic in up to 40% of cases [58] and, therefore, undetectable by karyotype analysis. Moreover, the ability to detect mosaicism is directly related to the number of analyzed cells, and the results obtained (2.3% in our case) might not always be accurate. More sensitive techniques (for instance SNP microarray) are required for a better assessment of low-grade mosaicism.

## 5. Conclusions

We found a slightly higher chromosome abnormality rate in sporadic first-trimester miscarriages through conventional karyotyping with QF-PCR as a complementary molecular technique to detect maternal cell contamination. Our results confirmed some previous findings that older women with spontaneous miscarriages have a higher rate of chromosomal abnormalities. The abnormality rate detected in first-trimester miscarriages can be increased upon associating karyotyping with molecular genetic techniques. Further large-scale studies comparing diagnostic results of different genetic techniques are required.

## Figures and Tables

**Table 1 genes-13-02246-t001:** Abnormal cases by chromosome abnormality type.

Chromosome Abnormality		*n*	%
Single trisomy		71	52.6
Double trisomy		3	2.2
Monosomy X		24	17.8
Mosaic abnormalities		3	2.2
Polyploidy		23	17.1
Structural abnormalities	Balanced	3	2.2
	Unbalanced	8	5.9
Total		135	100.0

**Table 2 genes-13-02246-t002:** Chromosome involvement in all single trisomies detected.

Single Trisomy	*n*	% of Single Trisomies	% of Chromosome Abnormalities *
Trisomy 2	1	1.4	0.7
Trisomy 7	1	1.4	0.7
Trisomy 8	3	4.2	2.2
Trisomy 9	3	4.2	2.2
Trisomy 10	1	1.4	0.7
Trisomy 12	1	1.4	0.7
Trisomy 13	4	5.6	3
Trisomy 14	2	2.8	1.5
Trisomy 15	6	8.5	4.4
Trisomy 16	23	32.4	17
Trisomy 18	7	9.9	5.2
Trisomy 21	10	14.1	7.4
Trisomy 22	9	12.7	6.7
Total	71	100.0	52.6

* round values.

**Table 3 genes-13-02246-t003:** Structural abnormalities detected.

Structural Abnormality	Type	Karyotype
Unbalanced	Terminal deletion	46,XX, del(9q)
		46,XX, del(8)(q24)
	Isochromosome	47,XX, +i(12)(p10)
	Unbalanced reciprocal translocation	45,XX, der(9)t(9;21)(p24.3;q22.1), −21
	Unbalanced Robertsonian translocation	46,XX, rob(13;14)(q10;q10), +14
		46,XY, rob(13;21)(q10;q10), +21
		46,XX, rob(14;15)(q10;q10), +15
		46,XY, rob(15;15)(q10;q10), +15
Balanced	Pericentric inversion Balanced Robertsonian translocation	46,XX, inv(2)(p11.2q13)
		46,XX, inv(10)(p11.2q21)
		45,XX, rob(14;21)(q10;q10)

**Table 4 genes-13-02246-t004:** Chromosomal abnormalities stratified by maternal age.

Age (Years)	Normal	Trisomy	Monosomy X	Polyploidy	Structural Abnormality	Mosaic	Total
<35	70	41 *	16	19	5	2	153
≥35	25	33 **	8	4	6	1	77
Total	95	74	24	23	11	3	230

* includes 1 case with double trisomy. ** includes 2 cases with double trisomy.

**Table 5 genes-13-02246-t005:** Relative frequency of chromosome abnormality types in different studies (all data adjusted to the terms of the present study, rounded values).

Study	Type of Chromosome Abnormality (%)				Abnormality Rate (% of Total Cases)
	Trisomies	Monosomy X	Polyploidy	Structural Abnormalities	
Present study	54.8	17.8	17.0	8.1	58.7
Hassold, 1980	46.4	24.0	22.2	4.3	46.3
Ljunger, 2005	55.3	9.4	14.5	6.9	61.0
Menasha A, 2005	67.7	13.7	13.0	4.9	42.8
Menasha B, 2005	73.2	6.8	14.1	3.9	65.8
Petracchi, 2008	55.8	14.2	16.1	4.8	44.7
Pylyp, 2017	63.5	7.5	22.0	7.0	50.1
Soler, 2017	69.1	9.8	14.5	5.2	70.3
Horiuchi, 2019	53.9	7.9	10.1	12.4	67.4
Gomez, 2020	61.3	10.4	15.0	13.4	64.3
Wu, 2021	65.8	10.9	11.3	5.7	53.2
Zhang, 2021	76.4	9.7	6.6	5.5	48.5

**Table 6 genes-13-02246-t006:** Relative frequency of the 6 most common single autosomal trisomies in previous studies and in the present series. All data adjusted to the terms of the present study.

Study	Most Common Trisomies and Respective % (of Single Autosomal Trisomies)					
Present study	T16 (32.4%)	T21 (14.1%)	T22 (12.7%)	T18 (9.9%)	T15 (8.5%)	T13 (5.6%)
Hassold, 1980	T16 (24.8%)	T22 (14.1%)	T21 (12.6%)	T15 (7.8)	T13 (4.9%)	T7 (3.88%)
Ljunger, 2005	T16 (43.2%)	T22 (9.9%)	T2 (7.4%)	T18 (6.2%)	T3, T13, T15, T21 (3.7%)	T18, T9, T14, T20 (2.5%)
Menasha A, 2005	T21 (3.9%)	T16 *	T22 *	T18 *	-	-
Menasha B, 2005	T16 (18.4%)	T22 *	T21 *	T15 *	-	-
Pylyp, 2017	T16 (18.3%)	T21 (14.8%)	T22 (13.4%)	T13 (10.3%)	T15 (9.7%)	T18 (8.3%)
Soler, 2017	T16 (18.73)	T22 (18.5%)	T15 (14.2%)	T21 (12.2%)	T13 (6.5%)	T18 (5.2%)
Horiuchi, 2019	T16 (25.5%)	T22 (21.3%)	T15 (17%)	T2, T13 (8.5%)	T21 (6.4%)	T12 (4.3)
Gomez, 2020	T16 (25.7%)	T22 (13.2%)	T21 (7.9%)	T13 (6.6%)	T18 (3.9%)	-
Wu, 2021	T16 (30%)	T22 (14.6%)	T21 (9.1%)	T13 (5.8%)	T8 (4.6%)	T7, T9 (3.6%)
Zhang, 2021	T16 (26.5%)	T22 (15.4%)	T14 (7.7%)	T4 (6%)	T18 (5.1%)	T8, T10, T15 (4.3%)

* Data values are not available.

## Data Availability

All data presented here are available from the authors upon reasonable request.

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
