# Peer review of "Cytogenetic Analysis of Sporadic First-Trimester Miscarriage Specimens Using Karyotyping and QF-PCR: A Retrospective Romanian Cohort Study"

_genes, 2022, doi:10.3390/genes13122246_

Round 1
Reviewer 1 Report
The authors describe their experience with karyotyping first-trimester miscarriage specimens in Romania and the various abnormalities they found, and they put their data in the context of data from other relevant studies.
The paper is beautifully written, clear and concise. The data are similarly presented in a clean and easy-to-understand format. It was really a pleasure to read and review.
I have some very minor suggestions which are listed below line by line:
Line 30: change "being the most detected ones" to "being the ones most commonly detected"
Line 42: QF-PCR-proven is better than QF-PCR proved
Lines 43-45: I think aside from P values, you don't need to report the percentage to the hundredths place. For example, 58.69% would be fine to report as 58.7%. I do think you should put a decimal for 23/135 - 17.0% however. Also, in Line 45, instead of saying "single one" for monosomy X, better would be "only one."
Line 69: X monosomies would be better as monosomy X
Line 99: South-Western could be changed to southwestern
Line 140: was should be were
Lines 152-153: 41.31% -> 41.3% and 58.69% -> 58.7%
Line 164: consisted of instead of consisted in
Line 175: insert comma after +12 so it is +12,+13 for ISCN nomenclature
Table 2: change 0.74 to 0.7%
Table 4: Poliploidy should be spelled Polyploidy
Line 216: 10-year study instead of 10-years study
Line 220: there are reported data instead of there is reported data
Line 221: Our findings compared to instead of related to
Table 5: Polyploidy instead of Poliploidy, and please remove the hundredths place decimals
Line 269: put in vitro in italics, since you put de novo in italics in line 66
Author Response
Dear Reviewer,
We are grateful for the amount of time and effort that you dedicated to providing feedback on our manuscript and are thankful for the insightful comments on and valuable improvements to our paper.
Please see the attachment with a point-by-point response to your comments and concerns, marked in blue.
We have made all efforts to address all the concerns you raised. The changes can be found in the track-changes revised manuscript.
We would like to express our thanks to the reviewers for the constructive and positive comments.
Best wishes,
The authors

Reviewer 2 Report
This research article written by Popescu-Hobeanu and colleagues titled “Cytogenetic analysis of sporadic first-trimester miscarriage specimens using karyotyping and QF-PCR: a retrospective Romanian cohort study” is interesting and well written. This study will contribute to increase our understanding of human miscarriages. However, the authors should address my concerns listed below before this can be accepted for publication.
1. This research has been conducted using human fetal tissues/cells. Therefore, this research project should have been approved by the Institutional Research board (IRB)? If it was approved, please write IRB number and approval date in methods section. If there is no such approval this paper should not be published.
2. For the benefit of general readers please include a brief description of the principle of QF-PCR in method section. People may confuse this technique with quantitative real time PCR sometimes refers to as qPCR.
3. This study shows that 58.69% of miscarriages are due to chromosome abnormalities. This does not mean that remaining 41.31% pregnancies are normal. Please discuss in discussion section the potential of other genetic and epigenetic alterations that that may contribute towards miscarriages apart from chromosome abnormalities.
Author Response

(The authors gave the same response as above.)
